# Copper and Cadmium Accumulation and Phytorextraction Potential of Native and Cultivated Plants Growing around a Copper Smelter

**Changming Dou [1], Hongbiao Cui [2,*], Wei Zhang [2], Wenli Yu [2], Xue Sheng [2] and Xuebo Zheng [3,*]**

[1] Anhui Provincial Academy of Eco-Environmental Science Research, Hefei 230061, China; doucm@sina.com
[2] School of Earth and Environment, Anhui University of Science and Technology, Huainan 232001, China
[3] Key Laboratory of Tobacco Biology and Processing, Ministry of Agriculture, Tobacco Research Institute of Chinese Academy of Agricultural Sciences, Qingdao 266101, China
* Correspondence: hbcui@aust.edu.cn (H.C.); zhengxuebo@caas.cn (X.Z.)

**Abstract:** Phytoextraction is a promising technology for remediating heavy metal-contaminated soil. Continuously screening potential plants is important for enhancing the efficiency of remediation. In this study, fourteen local native plant species and four cultivated plant species, along with their paired soils, were collected from around a copper smelter. The characteristics of soil pollution were evaluated using contaminant factors (CF) and a geoaccumulation index (Igeo). The phytoextraction potential of plants was investigated using the translocation factor (TF) and bioconcentration factor (BCF). The soils around the smelter were very acidic, with a mean pH of 5.01. The CF for copper and cadmium were 8.67–32.3 and 5.45–44.2, and the Igeo values for copper and cadmium were 2.43–4.43 and −0.12–2.29, respectively, indicating that the level of soil contamination was moderate to severe. The copper concentrations in the root (357 mg/kg), shoot (219 mg/kg), and leaf (269 mg/kg) of *Elsholtzia splendens* Nakai were higher than that in the other species. The cadmium in the shoot (32.2 mg/kg) and leaf (18.5 mg/kg) of *Sedum plumbizincicola* was the highest, and *Phytolacca acinosa* Roxb. had the highest cadmium level (20 mg/kg) in the root. Soil total and CaCl$_2$-extractable copper and cadmium were positively correlated with copper and cadmium in the plant roots, respectively. The results of TF and BCF for copper and cadmium suggested that the accumulation and translocation capacities for cadmium were higher than those of copper in the eighteen plant species. Although not all plants met the criteria of being hyperaccumulators, *Sedum plumbizincicola*, *Mosla chinensis* Maxim, and *Elsholtzia splendens* Nakai showed the most potential as candidates for the phytoextraction of copper and cadmium contaminated soils, as indicated by their TF and BCF values.

**Keywords:** heavy metals; phytoremediation; environmental risk; accumulation capacities; hyperaccumulator

## 1. Introduction

Heavy metals (HMs) are characterized by a specific weight of more than 5.0 or a density over 4.5 g/cm$^3$, which cannot be degraded by microorganisms and pose significant hazards to plant growth and human health [1]. With the rapid urbanization and industrialization, large amounts of HMs have been accumulated in soil and water due to activities such as smelting, mining, fertilization, and irrigation. Currently, soil contamination by HMs is a global issue, particularly in developing countries [2]. According to a survey by Chinese government, 16.1% of soil sampling points exceeded the environmental quality standard and 82.8% of these points were polluted by cadmium, mercury, arsenic, copper, lead, etc. [3]. Moreover, the survey showed that the majority of the polluted soils were located near heavily polluting enterprises, industrial wastelands, smelters, and mining areas. HMs in soil and water can accumulate in crops and fish, posing a potential hazard to human health through the food chain.

Compared with the traditional physical and chemical remediation techniques, phytoremediation is a promising method for remediating soils contaminated with HMs, because this technique is advantageous due to its cost-effectiveness and environmentally friendly nature [4,5]. This technique can purify the environment through phytoextraction, phytostabilization, phytovolatilization, phytofiltration, and phytotransformation. Of these methods, the phytoextraction is the most widely used [6]. Phytoextraction can effectively remove soil HMs by harvesting the plants that can accumulate large amounts of HMs in their aboveground parts [7,8]. For phytoextraction, promising plants are characterized by high biomass, high growth rate, and the ability to accumulate HMs [4,9]. But plant genotype, soil physical, and chemical characteristics (pH, organic matter, cation exchange capacity, etc.); climatic conditions (temperature, rainfall); and the availability of HMs all influence the efficiency of phytoextraction [9,10]. Therefore, identifying and selecting appropriate plants that tolerate or have hyper-accumulate capacities for HMs is the key to successful phytoextraction [9,11].

Previous studies have shown that both wild and cultivated plants were extensively utilized for phytoextraction [11,12]. As for artificially cultivated restoration plant species, they mainly consist of hyper-accumulators, such as *Sedum plumbizincicola* (a cadmium hyper-accumulator), *Pteris vittata* L. (an arsenic hyper-accumulator), and *Elsholtzia splendens* Nakai (a copper-tolerant plant). These species may be replaced by local native plant species in the polluted area once human intervention is withdrawn [13–16]. On the contrary, native wild plants can grow robustly after undergoing long-term natural selection and can withstand a series of natural disasters, such as local extreme climate conditions, pests, and diseases [11,17]. Moreover, some studies have reported that the phytoextraction efficiency for HMs in artificially cultivated hyper-accumulators was lower in field experiments compared to pot experiments [18–20]. For example, the concentrations of cadmium in *Sedum plumbizincicola* grown in a pot soil (total cadmium, 1.25 mg/kg) were 0.44 times higher than those grown in a cadmium-contaminated field (total cadmium, 1.25 mg/kg) [21]. However, the copper contents in the shoots of *Elsholtzia splendens* Nakai under field conditions were 24 times higher than in those grown under greenhouse conditions [22]. The phytoextraction efficiency for hyper-accumulators depends on the environmental conditions. Thus, the further assessment of the phytoextraction potential for both native and cultivated plants is needed when they are all grown under similar environmental conditions.

The goal of this study was to screen the phytoextraction potential of plants species by comparing native and cultivated plants growing around a copper smelter. In this study, we collected fourteen local native plant species and four cultivated plant species. The cultivated plants were grown in a demonstration zone and included *Pennisetum sinese* Roxb. (an energy plant), *Sedum plumbizincicola* (a cadmium hyper-accumulator), *Elsholtzia splendens* Nakai (a copper-tolerant plant), and *Canna indica* L. (a landscape plant). Objectives included: (1) investigating the pollution level of the soils using contaminant factors (CF) and the geoaccumulation index (Igeo); (2) exploring the concentration of copper and cadmium in different plant tissues; and (3) determining the phytoextraction potential of plants based on the translocation factor (TF) and bioconcentration factor (BCF).

## 2. Materials and Methods

### 2.1. Site Description

The copper smelter is located in Guixi city, China (Figure 1), and was established in 1985. The long-term smelter activities have released significant amounts of HMs into the surrounding area, resulting in severe crop pollution and posing high health risks to local residents [23]. The primary pollutants in the region include copper and cadmium, which originate from the wastewater, atmospheric deposition, and leachate from solid waste. In previous studies, there was a high atmospheric deposition of copper and cadmium in the northwest and southeast areas near the smelter. Additionally, the soil in these areas was found to be highly acidic and heavily contaminated by copper and cadmium [24,25].

Therefore, the paired plant and soil samples were collected from these two directions near the smelter.

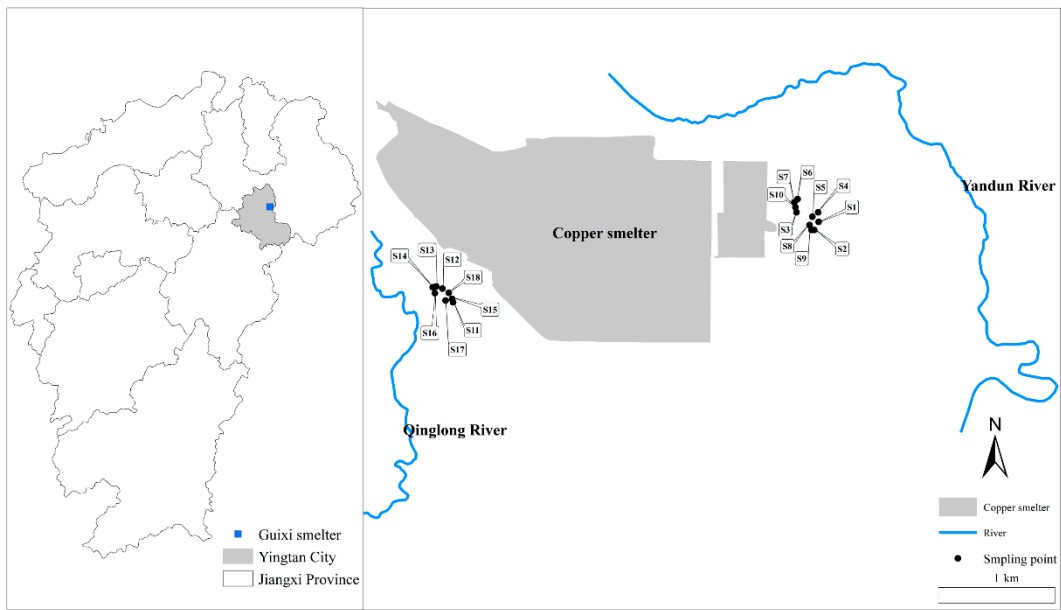

**Figure 1.** Map of plant sample collection sites around the Guixi smelter.

### 2.2. Samples Collection and Chemical Analysis

Eighteen plant species were sampled, and the codes (S1–S18), latitude, and longitude are listed in Table S1 and Figure 1. Among the plants, *Pennisetum sinese* Roxb. (S11), *Sedum plumbizincicola* (S16), *Elsholtzia splendens* Nakai (S13), and *Canna indica* L. (S14) were collected from a demonstration plot for the remediation of HM-contaminated soil, while the others were native wild plants. The paired soil samples were also collected from the rhizosphere of the plant, and both the plant and soil samples were replicated in triplicate for each site.

The tiny ash from plants surfaces was washed successively with tap water, 10% $HNO_3$, and deionized water. They were then divided into roots, shoots (without leaf), and leaves, respectively. All plant samples were dried at 85 °C until constant weight in an oven, and then ground to a powder. Copper and cadmium in plant tissues were analyzed using atomic absorption spectrometry with graphite furnace (AAS-GF) after digestion with 5 mL of $HNO_3$ and 1 mL of $HClO_4$ on a hot plate. A plant reference (GBW10010, rice) was analyzed to ensure the quality control.

Soil samples were air-dried indoors for 14 days. One portion of the soil was sieved using a 2 mm sieve and the other portion was sieved using a 0.15 mm sieve. The soil samples that passed through a 2 mm sieve were used to determine soil pH using a pH electrode. Available copper and cadmium were determined after extraction with 0.01 mol/L $CaCl_2$ at a ratio of 1:5 (solid to liquid) [24]. Total copper and cadmium in soils that passed through 0.15 mm sieve were determined using AAS-GF after digestion with a mixture of acids on a hot plate [26]. Meanwhile, a soil reference material (GBW07405) was analyzed for quality control.

### 2.3. Data Analysis

2.3.1. Contaminant Factors (CF)

The CF can reflect the pollution level of soil, and it was calculated using Equation (1) [27]. The CF consisted of four levels: low (CF < 1), moderate ($1 \leq$ CF < 3), considerable ($3 \leq$ CF < 6), and very high ($6 \leq$ CF) contamination [28].

$$CF = C_{metal}/C_{background} \tag{1}$$

$C_{metal}$ (mg/kg) represents the metal contents in soil samples; $C_{background}$ (mg/kg) represents the metal contents in background soil (Cu 20.3 mg/kg; Cd 0.11 mg/kg).

### 2.3.2. Geoaccumulation Index (Igeo)

Igeo can be calculated using Equation (2) and can quantify the extent of metal pollution for the studied soil [27]. The Igeo is divided into seven grades (Table 1) [29].

$$Igeo = \log_2(C_n/1.5B_n) \tag{2}$$

where $C_n$ and $B_n$ represent the metal content (n) in the soil sample and the background content of the metal (n) (Cu 20.3 mg/kg; Cd 0.11 mg/kg). Factor 1.5 is the background matrix correction factor due to lithospheric effects.

**Table 1.** Pollution level and values of Igeo.

| Grade Level | Values of Igeo |
|---|---|
| Grade 0: unpolluted | Igeo $\leq$ 0 |
| Grade 1: unpolluted to moderately polluted | 0 < Igeo < 1 |
| Grade 2: moderately polluted | 1 $\leq$ Igeo < 2 |
| Grade 3: moderately to heavily polluted | 2 $\leq$ Igeo < 3 |
| Grade 4: heavily polluted | 3 $\leq$ Igeo < 4 |
| Grade 5: heavily to extremely polluted | 4 $\leq$ Igeo < 5 |
| Grade 6: extremely polluted | 5 $\leq$ Igeo |

### 2.3.3. Translocation Factor (TF)

The TF can be used to assess the capacity of plants to translocate metals from the root to shoot:

$$TF = C_{shoot}/C_{root} \tag{3}$$

where the $C_{shoot}$ and $C_{root}$ are the concentrations of metals in shoot and root, respectively. If the TF > 1, it represents the high capacity for removing HMs from soil [6].

### 2.3.4. Bioconcentration Factor (BCF)

BCF can be used to evaluate the efficiency of plants adsorbing metals from the soil to their aboveground parts. It can be calculated using Equation (4) [30]. When the BCF >1, it suggests the high potential for plant phytoextraction [31].

$$BCF = C_{shoot}/C_{soil} \tag{4}$$

where the $C_{shoot}$ and $C_{soil}$ are the contents of metals in shoot and soil samples, respectively.

### 2.4. Statistical Analyses

All data were analyzed using SPSS 28.0 software (IBM, Armonk, NY, USA). Each datapoint is the mean ± standard error. Comparisons of datasets were performed using a one-way analysis of variance (ANOVA) test at a significance level of $p < 0.05$.

## 3. Results and Discussion

### 3.1. Total and Available Copper and Cadmium in Soils

The soil pH in the plant rhizosphere around the smelter ranged from 4.54 to 5.85 (Table 2). The mean pH was 5.01, and over 50% of the soil values were less than 5.0, indicating that the soil was very acidic. This may be due to the fact that the soils were derived from Quaternary red clay, which is classified as *Ultisols* [24]. These soils were affected by acid rain with a pH of 3.09–4.50 [32]. Total copper and total cadmium in soil ranged from 176 to 655 mg/kg and from 0.60 to 4.87 mg/kg, respectively. The CFs for copper and cadmium were 8.67–32.3 and 5.45–44.2 (Figure S1), respectively, which were significantly higher than 6 (except for *Mosla chinensis* Maxim). This indicates that the

study area was severely contaminated due to smelter activities [27,28,33]. Furthermore, the concentrations of soil copper and cadmium were 4.52–12.1 and 1.0–15.2 times higher than the recommended screening values for copper and cadmium (GB 15618-2018, pH < 5.5), respectively. Meanwhile, the soils around the smelter were also severely contaminated compared to the clean soil located about 16.4 km away (copper 33.3 mg/kg, cadmium 0.368 mg/kg; Table 2) [34]. Moreover, the $I_{geo}$ values were 2.43–4.43 for copper and −0.12–2.29 for cadmium (Figure S2). The $I_{geo}$ values indicated moderate to heavy pollution for copper and cadmium, except for the soil from *Mosla chinensis* Maxim.

**Table 2.** Soil pH and total and available copper (mg/kg) and cadmium (mg/kg) around the Guixi smelter.

| Species of Plants | Soil pH | Total Cu | Available Cu | Total Cd | Available Cd |
|---|---|---|---|---|---|
| *Oenothera biennis* L. | 5.02 ± 0.08 bcd | 290 ± 5 hi | 41.3 ± 3.5 bc | 1.63 ± 0.05 ghi | 0.52 ± 0.27 h |
| *Cyclosorus interruptus* | 4.92 ± 0.05 bcd | 430 ± 42 cd | 64.7 ± 2.9 abc | 2.73 ± 0.08 c | 1.31 ± 0.03 c |
| *Verbena officinalis* L. | 5.03 ± 0.03 bcd | 250 ± 26 i | 35.6 ± 4.1 c | 2.03 ± 0.16 ef | 1.02 ± 0.08 d |
| *Xanthium strumarium* L. | 4.93 ± 0.04 bcd | 333 ± 34 efgh | 64.5 ± 4.3 abc | 2.26 ± 0.16 de | 1.03 ± 0.06 d |
| *Solidago canadensis* L. | 5.14 ± 0.02 abcd | 331 ± 43 efgh | 57.1 ± 4.3 abc | 1.64 ± 0.06 ghi | 0.75 ± 0.06 fg |
| *Saccharum arundinaceum* Retz. | 5.09 ± 0.03 abcd | 319 ± 28 fghi | 51.7 ± 6.8 abc | 1.92 ± 0.16 fg | 0.96 ± 0.08 de |
| *Pteris multifida* Poir. | 4.58 ± 0.03 d | 559 ± 44 b | 99.7 ± 11.8 a | 3.18 ± 0.24 b | 1.59 ± 0.12 b |
| *Pteris vittata* L. | 5.44 ± 0.04 abc | 412 ± 14 cd | 45.2 ± 7.2 abc | 1.44 ± 0.12 hij | 0.48 ± 0.03 h |
| *Phytolacca acinosa* Roxb. | 4.8 ± 0.03 bcd | 176 ± 20 j | 29.8 ± 2.6 c | 4.87 ± 0.33 a | 2.43 ± 0.17 a |
| *Artemisia sieversiana* Ehrhart ex Willd. | 4.68 ± 0.02 cd | 394 ± 14 cdef | 68.7 ± 2 abc | 1.7 ± 0.11 gh | 0.87 ± 0.04 def |
| *Pennisetum sinese* Roxb. | 4.65 ± 0.04 cd | 441 ± 39 cd | 76.5 ± 3.4 abc | 1.95 ± 0.06 efg | 0.97 ± 0.04 de |
| *Lophatherum gracile* Brongn. | 4.81 ± 0.04 bcd | 399 ± 23 cde | 65 ± 6 abc | 1.36 ± 0.06 ij | 0.68 ± 0.04 fgh |
| *Elsholtzia splendens* Nakai | 4.74 ± 0.03 bcd | 586 ± 51 ab | 95.7 ± 11.3 ab | 1.17 ± 0.18 j | 0.57 ± 0.07 gh |
| *Canna indica* L. | 5.85 ± 1.41 a | 655 ± 33 a | 76.6 ± 5.1 abc | 1.61 ± 0.06 ghi | 0.77 ± 0.05 efg |
| *Aster subulatus* Michx. | 5.29 ± 0.03 abcd | 370 ± 25 defg | 57.6 ± 9.8 abc | 1.47 ± 0.06 hij | 0.69 ± 0.04 fgh |
| *Sedum plumbizincicola* | 4.55 ± 0.02 d | 462 ± 56 c | 86.4 ± 5.4 abc | 2.51 ± 0.12 cd | 1.3 ± 0.01 c |
| *Mosla chinensis* Maxim | 5.54 ± 0.04 ab | 311 ± 17 ghi | 37.3 ± 3.5 c | 0.6 ± 0.08 k | 0.25 ± 0.03 i |
| *Vetiveria zizanioides* L. | 5.15 ± 0.04 abcd | 600 ± 14 ab | 79.9 ± 11.9 abc | 3.18 ± 0.08 b | 1.49 ± 0.1 |
| Mean value | 5.01 | 406 | | 2.07 | |
| Clean soil * | | 33.3 | n.d. | 0.368 | 0.015 |
| Background value | | 20.3 | | 0.11 | |
| Soil screening value (GB 15618-2018, pH ≤ 5.5) | | 50 | | 0.3 | |

Note: * the data of the clean soil was obtained from the study by Cui et al. [34]. Mean (*n* = 3) and standard error followed by different letters indicates significant differences (*p* < 0.05). n.d., not detectable.

The above results suggest that the soils in the study area are significantly more contaminated than clean soil. This contamination may be attributed to the smelter activities, including the atmospheric deposition, wastewater discharge, leaching solutions from solid waste, etc. Furthermore, the $CaCl_2$ extractable copper and cadmium concentrations were also significantly higher than those found in the clean soil, suggesting that these metals may be easily transported into crops [35]. Our recent study showed that local residents were exposed to high health risks due to the presence of heavy metals (HMs) in rice, vegetables, eggs, etc. [23]. Therefore, remediating the contaminated soils and controlling the potential risk to human health are meaningful.

### 3.2. Accumulation of Copper and Cadmium in Plants

To mitigate the impact of extreme values on the assessment of metal accumulation in plants, the top and bottom 25% of copper and cadmium concentrations were excluded. The distribution of copper and cadmium accumulation in different plant tissues are shown in Figure 2. Similar to the concentrations of metals in soil, the mean metal contents in the roots, shoots, and leaves of plants all followed the order of copper > cadmium. Moreover, the mean copper contents in plants followed the root > leaf > shoot order, while

cadmium followed the root > shoot > leaf order. The results indicated the significant variations of metal uptake capacities among different parts of plant species. For instance, the copper concentrations in the roots (47.2–156 mg/kg) were significantly higher than those of cadmium (1.81–3.73 mg/kg). Meanwhile, the copper concentration in the plant leaves ranged from 32.3 to 74.4 mg/kg, which fell within the phytotoxic level (20–100 mg/kg) [36]. This may be the main reason for the decline in crop yield and the metal accumulation in crops in the local area.

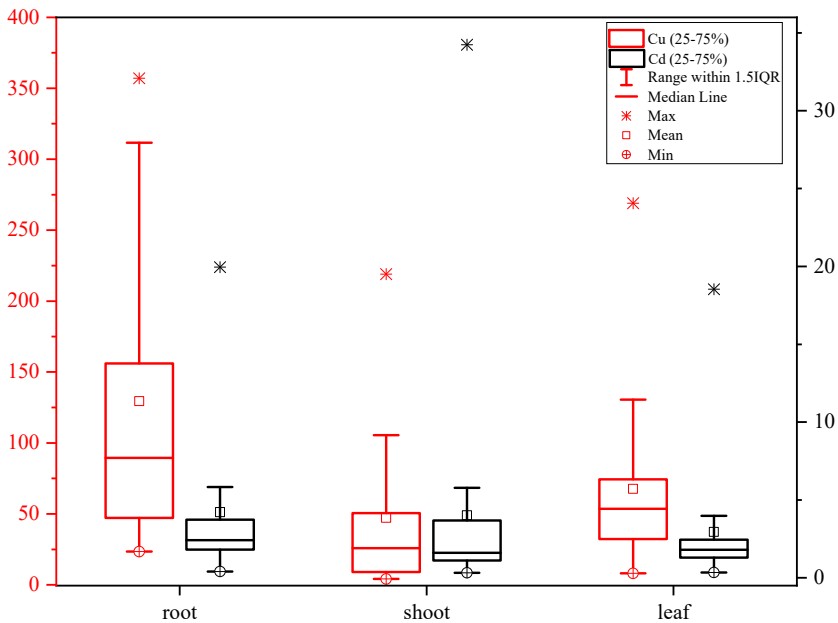

**Figure 2.** Concentrations of copper and cadmium in different parts of the plants around the Guixi smelter.

Furthermore, the specific concentrations of copper and cadmium in the root, shoot, and leaf are provided in Table S2. The highest copper contents in the root, shoot, and leaf were found in *Elsholtzia splendens* Nakai, with the values of 357 mg/kg, 219 mg/kg, and 269 mg/kg, respectively. The lowest copper concentrations in the root (23.5 mg/kg) and leaf (8.07 mg/kg) were both found in *Saccharum arundinaceum* Retz. The shoot of *Solidago canadensis* L. had the lowest copper concentration (4.18 mg/kg). The highest concentration of cadmium in the root was found in *Phytolacca acinosa* Roxb. (20.0 mg/kg), followed by *Sedum plumbizincicola* (16.4 mg/kg). The highest levels of cadmium were found in the shoot (32.2 mg/kg) and leaf (18.5 mg/kg) of *Sedum plumbizincicola*, while the lowest levels were found in the shoot and leaf of *Solidago canadensis* L. (0.79 mg/kg) and *Lophatherum gracile* Brongn. (0.34 mg/kg), respectively. Moreover, the copper concentrations in the plant root were significantly lower than those in their paired soils. However, 72.2% of the cadmium concentrations in plant roots were higher than those in the associated rhizosphere soil. Soil total and available copper were positively correlated with the copper concentrations in plant roots ($p < 0.01$) (Figure 3). Similarly, the total and available cadmium in the soil were positively correlated with the cadmium concentrations in the plant roots ($p < 0.01$) (Figure 3). This indicated that the metal accumulation in plants was significantly influenced by total and available metal concentrations in the soil.

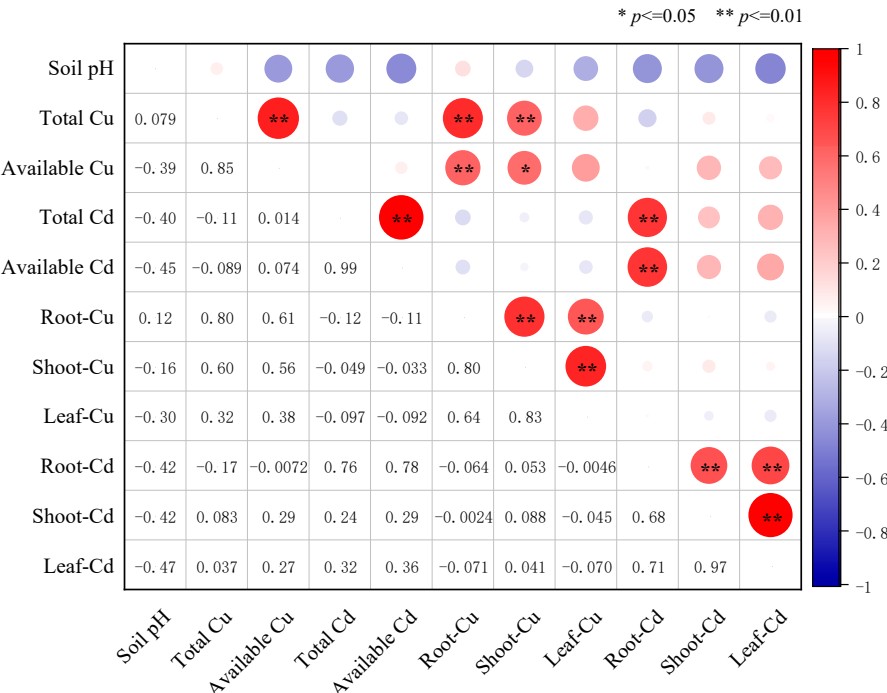

**Figure 3.** Correlation coefficients of total soil and available copper and cadmium and metal accumulation by plants.

In general, the *Elsholtzia splendens* and *Sedum plumbizincicola* had a higher potential for accumulating copper and cadmium, respectively, compared to other plants. For the copper-tolerant plant, copper in *Elsholtzia splendens* is bound to the cell wall and histidine, and its detoxification occurs the complexation with nitrogen/oxygen and S ligands [37,38]. Some specific proteins, including pABCB28, SpMTP5, SpNRAMP5, SpHMA2, and SpHMA3, are involved in the translocation and detoxification of cadmium [39,40]. These proteins play important roles in the hyper-accumulation capacity for cadmium in *Sedum plumbizincicola*. Specifically, SpHMA3 is responsible for transporting cadmium and localizing it to the tonoplast. It plays a crucial role in the detoxification of cadmium in the shoots of *Sedum plumbizincicola* by sequestering cadmium into the vacuoles [41]. However, not all eighteen species of plants met the threshold value for copper (1000 mg/kg) and cadmium (100 mg/kg) hyperaccumulation [42,43]. For example, the concentration of cadmium in the shoot (34.2 mg/kg) of *Sedum plumbizincicola* in this study was significantly lower than the average value of cadmium (170–172 mg/kg) reported by Wang et al. [18]. This may be due to the following reasons. First, the metal accumulation in plants depends on the total and available metal in the soil. For example, the cadmium content in the shoots of *Sedum plumbizincicola* reached 100–540 mg/kg when the total cadmium in the soil was 8.68–16.9 mg/kg, as reported by Li et al. [44]. This concentration was 1.78–3.47 times higher than the highest soil cadmium observed in this study. Second, the climatic conditions including the rainfall deposition and temperature also significantly influenced the behavior of metal uptake [45]. Finally, the source of HMs and their bioavailability also significantly affected the metal accumulation in plants. In particular, the atmospheric deposition of copper and cadmium in the study area ranged from 106–1369 mg/m$^2$ and 2.3–10.3 mg/m$^2$, respectively, which were the main sources of the soil metal [23]. Meanwhile, the 84% and 87% of atmospheric the wet deposition of copper and cadmium were found in ionic speciation, making them more prone to accumulation in the above-ground parts of plants [23,46]. This may be the reason for higher copper concentrations in plant leaves compared to shoots (Figure 2). Similarly, our recent study showed that the accumulation of copper and cadmium in the new leaves of campour at high deposition points was much higher compared to that at low deposition points [25].

### 3.3. TF and BCF of Plants

The TF values of copper and cadmium among the plants are shown in Figure 4. *Elsholtzia splendens* Nakai had the highest TF for copper (0.62), while *Solidago canadensis* L., *Artemisia sieversiana* Ehrhart ex Willd. and *Canna indica* L. had the lowest TF for copper (0.06–0.09). However, the $TF_{Cu}$ values for all the eighteen plants were less than 1, indicating a low accumulation capacity for copper. Regarding cadmium, *Mosla chinensis* Maxim had the highest $TF_{Cd}$ (3.39), followed by *Sedum plumbizincicola* (2.12), *Lophatherum gracile* Brongn. (1.54), *Pteris vittata* L. (1.39), *Cyclosorus interruptus* (1.20), *Saccharum arundinaceum* Retz. (1.18), *Xanthium strumarium* L. (1.15), and *Artemisia sieversiana* Ehrhart ex Willd. (1.03). The $TF_{Cd}$ values for the other ten plant species were all less than 1, suggesting a low capacity for cadmium translocation.

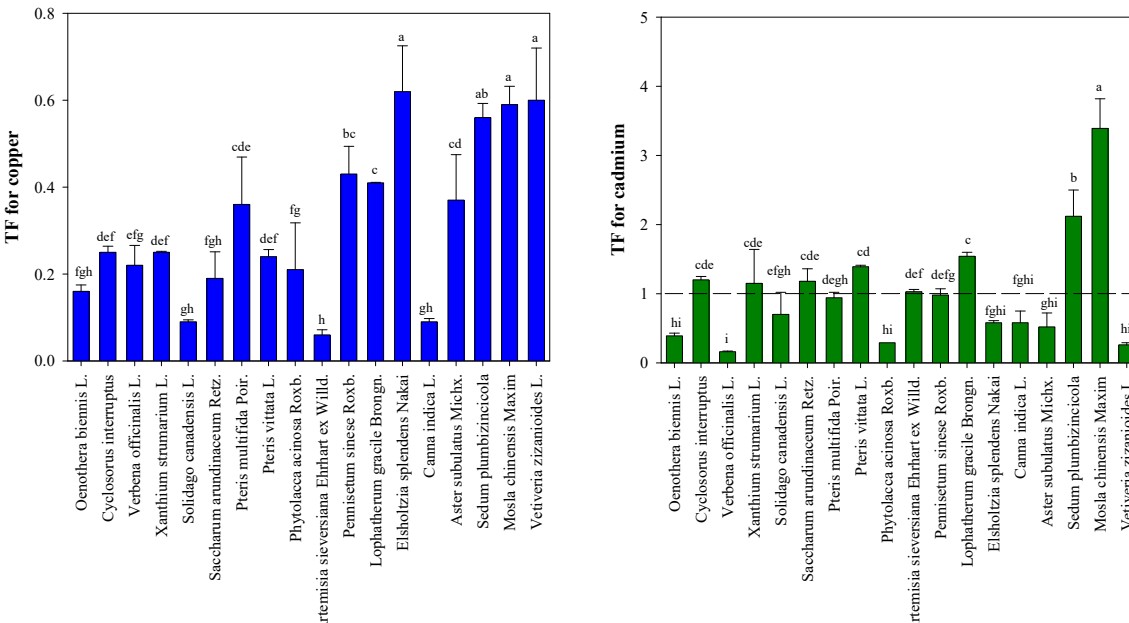

**Figure 4.** TF values of copper and cadmium for eighteen plant species. Mean (*n* = 3) and standard errors followed by different letters above the columns indicated significant difference at *p* < 0.05.

In addition to TF, the BCF values for copper and cadmium by the plants were also evaluated. The BCF values of copper ranged from 0.01 to 0.38 (Figure 5), and *Elsholtzia splendens* Nakai exhibited the highest $BCF_{Cu}$. The $BCF_{Cu}$ values for plants in this study were all lower than 1, suggesting a low capacity to accumulate copper into plant tissues for the eighteen plant species. The BCF values of cadmium ranged from 0.16 to 13.7. Among the plants tested, *Sedum plumbizincicola* exhibited the highest $BCF_{Cd}$ (13.7), which was 5.11 times higher than the second highest $BCF_{Cd}$ observed in *Mosla chinensis* Maxim (2.24). In contrast to $BCF_{Cu}$, the $BCF_{Cd}$ values in twelve of the eighteen plants were higher than 1, indicating that the plants had higher cadmium accumulation capacities compared to copper. Moreover, the values of $BCF_{Cd}$ for all the plants were higher than those of copper. Similarly, Wu et al. reported that the BCF values for HMs in plants followed the order of cadmium > zinc > copper > nickel > lead > chromium [47]. The higher accumulation capacity for cadmium by the plants may be due to the higher bioavailability of cadmium than that of copper, which was more easily adsorbed by plants [48].

It has been noted that BCF and TF values are more accurate in reflecting the metal accumulation and translocation capacities of different plants than the original metal concentrations in plants [49]. Especially for plants with high capacities to adsorb metals from the soil into their above-ground parts, the metal BCF and TF values were higher than 1 [36]. Consequently, we listed the top six plant species that have the potential for the phytoextraction of copper and cadmium (Table 3). It was clearly indicated that *Elsholtzia*

*splendens* Nakai had the highest capacities for copper accumulation among all the plant species. *Sedum plumbizincicola* and *Mosla chinensis* Maxim had high cadmium accumulation capacities and moderate copper accumulation capacity. Therefore, the three plant species can be considered potential candidates for phytoextraction in this study.

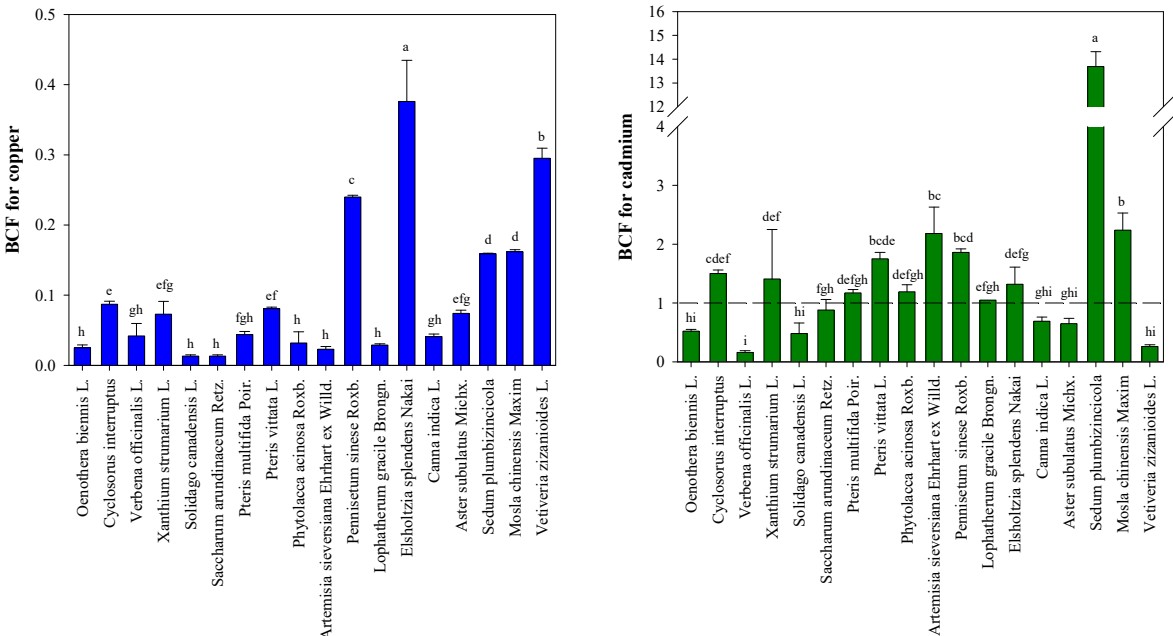

**Figure 5.** BCF values of copper and cadmium for eighteen plant species. Mean (*n* = 3) and standard error followed by different letters above the columns indicated significant difference at *p* < 0.05.

**Table 3.** The top six TF and BCF of Cu and Cd for plant species.

| TF$_{Cu}$ | BCF$_{Cu}$ | TF$_{Cd}$ | BCF$_{Cd}$ |
|---|---|---|---|
| *Elsholtzia splendens* Nakai (0.62) | *Elsholtzia splendens* Nakai (0.38) | *Mosla chinensis* Maxim (3.39) | *Sedum plumbizincicola* (13.7) |
| *Vetiveria zizanioides* L. (0.60) | *Vetiveria zizanioides* L. (0.30) | *Sedum plumbizincicola* (2.12) | *Mosla chinensis* Maxim (2.24) |
| *Mosla chinensis* Maxim (0.59) | *Pennisetum sinese* Roxb. (0.24) | *Lophatherum gracile* Brongn. (1.54) | *Artemisia sieversiana* Ehrhart ex Willd. (2.18) |
| *Sedum plumbizincicola* (0.56) | *Mosla chinensis* Maxim (0.16) | *Pteris vittata* L. (1.39) | *Pennisetum sinese* Roxb. (1.86) |
| *Pennisetum sinese* Roxb. (0.43) | *Sedum plumbizincicola* (0.16) | *Cyclosorus interruptus* (1.20) | *Pteris vittata* L. (1.75) |
| *Lophatherum gracile* Brongn. (0.41) | *Cyclosorus interruptus* (0.09) | *Saccharum arundinaceum* Retz. (1.18) | *Cyclosrus interruptus* (1.50) |

## 3.4. Phytoextraction Potential Evaluation and Environmental Implication

The selection of appropriate plant species is the key for the phytoextraction of soil polluted with HMs [50,51]. The ideal plant should have high/hyper metal accumulation capacities and considerable biomass, and it should be able to quickly remove HMs from soils [4]. This study did not find any heavy metal hyperaccumulators, but *Elsholtzia splendens* Nakai, *Sedum plumbizincicola*, and *Mosla chinensis* Maxim exhibited high BCF and TF for copper and cadmium. Similar to the majority of previous studies, this study also neglected to consider the biomass of plants, thus the removal efficiency for HMs cannot be calculated [11,37,52]. This is primarily because of the challenge in accurately calculating the biomass (kg/ha) of plants during field investigations. Nevertheless, our study can provide theoretical guidance for phytoextraction, based on the capacities of metal accumulation and translocation.

The best candidates for phytoextraction should be evaluated based on their efficiency in removing heavy metals (HMs), as well as the concentration of metals in plants and their biomass. This evaluation considers the following four aspects. First, the most important factor for phytoextraction is the total removal efficiency of HMs from the polluted soil by the plants. However, the biomass of the majority of the hyper-accumulators was at a low level, which restricted the removal efficiency for HMs [4,6]. For example, the biomass of *Sedum plumbizincicola* ranged from 1.38 to 6.33 t/ha (dry weight), which was signifi-

cantly lower than that of *Pennisetum sinese* Roxb. (39 t/ha), maize (8.7–45 t/ha), and reed (13.9 t/ha) [24,53,54]. Second, the economic cost, including the planting, management, and safe disposal of harvested biomass, should also be taken into considered for phytoextraction. Especially for the hyper-accumulators with low biomass and large amounts of HMs, most of them have no potential economic value and require additional investment in incinerators or landfills [55,56]. Recently, there has been advocacy for using energy plants such as *Salix*, *Populus*, *Miscanthus*, and *Arundo* to remediate soils contaminated with HMs. These plants have high biomass and fast-growing rate, making them suitable for this purpose. The biomass can be sold to biomass power plants and used to generate electricity [57–60]. Meanwhile, they can effectively remove HMs from soil due to their high biomass and moderate metal accumulation capacities. Our recent demonstration project also indicated that *Pennisetum sinese* Roxb. can remove 6142 g/ha of copper and 138 g/ha of cadmium. Additionally, it can generate a profit of 2604–2791 USD/ha in heavily contaminated soil. Furthermore, it can be used as feed for livestock in lightly contaminated soils (Figure 6). Third, the potential ecological benefits from the phytoextraction process were neglected in most cases. For example, some plants may require a significant amount of water and fertilization, which can lead to a decrease in groundwater levels and soil quality in the short term. Even some plants are considered alien species, which can potentially cause ecological disasters. Meanwhile, it is important to consider the suitability of plants for the local climate. Some tall plants, including *Pennisetum sinese* Roxb. and *reed*, can provide habitats for various organisms, including insects, birds, and snakes. For instance, *Sedum plumbizincicola* cannot survive at temperature below 273.15 K or above 303.15 K. Forth, the environmental aesthetic benefits should be taken into account for the phytoextraction. The polluted soils or mining wastelands around large cities could be redeveloped into parks. We could select restorative plants that have beautiful flowers or are suitable for large-scale planting. These plants could be planted alone or mixed with other plants to establish an ecological park.

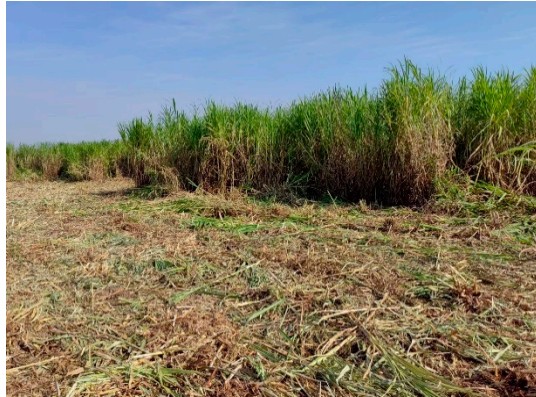 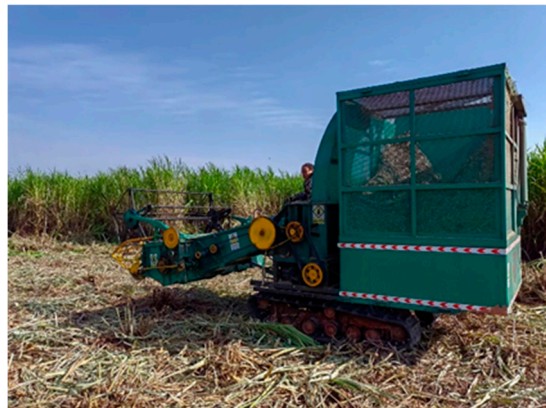

**Figure 6.** Demonstration project of soil remediation by *Pennisetum sinese* Roxb.

In general, *Sedum plumbizincicola*, *Mosla chinensis* Maxim, and *Elsholtzia splendens* Nakai were the most promising candidates for the phytoremediation of the polluted soils, based on their metal accumulation and translocation capacities. Nevertheless, the efficiency of metal removal should be further studied through pot or field experiments. In the future, it is essential to investigate and screen native or cultivated plants with high biomass, valuable properties, moderate HMs absorption capacity, and high metal removal efficiencies. Our previous studies had noted that the total removal efficiency of copper for *Pennisetum sinese* Roxb was 1.50 times higher than that of *Elsholtzia splendens* Nakai, and the removal efficiency of cadmium was similar to that of *Sedum plumbizincicola* [24]. Therefore, in the future, more efforts should be made to transform plants with a high biomass into fast-growing, valuable hyper-accumulators using transgenic technology. This will enable the rapid removal of heavy metals from the soil.

## 4. Conclusions

This study demonstrated that the soils around the smelter were moderately to heavily polluted with copper and cadmium and had a low soil pH. *Elsholtzia splendens* Nakai had higher copper concentrations in the roots, shoots, and leaves than the other plant species. The highest levels of cadmium were found in the shoots and leaves of *Sedum plumbizincicola*, while the highest concentration of cadmium in the roots was observed in *Phytolacca acinosa* Roxb. The eighteen plant species had higher accumulation and translocation capacities for cadmium compared to copper, as indicated by the TF and BCF. However, none of them met the criteria to be classified as hyperaccumulators. *Elsholtzia splendens* Nakai, *Sedum plumbizincicola,* and *Mosla chinensis* Maxim were the most promising candidates for the phytoextraction of copper and cadmium in the polluted soils. In the future, further studies are needed to examine the biomass, total removal efficiency, and economic benefits for these three plant species.

**Supplementary Materials:** The following supporting information can be downloaded at: https://www.mdpi.com/article/10.3390/agronomy13122874/s1, Table S1: The codes, latitude, and longitude of plants with paired soils; Table S2: Concentrations of copper (mg/kg) and cadmium (mg/kg) in roots, shoots, and leaves; Figure S1: The CF values for copper and cadmium around the copper smelter; Figure S2: The Igeo values for copper and cadmium around the copper smelter.

**Author Contributions:** Conceptualization, writing—original draft, C.D., H.C., W.Z. and X.Z.; data curation, supervision, H.C., W.Z. and C.D.; formal analysis, supervision, W.Z., C.D. and H.C.; investigation, resources, C.D., H.C. and X.Z.; validation, visualization, W.Y., H.C. and X.Z.; software, investigation, C.D., H.C., W.Z., X.S. and X.Z.; All authors have read and agreed to the published version of the manuscript.

**Funding:** This research was supported by the Anhui Province Natural Science Foundation (2208085MD87) and the Natural Science Foundation of Universities of Anhui Province (KJ2020ZD35).

**Institutional Review Board Statement:** Not applicable.

**Informed Consent Statement:** Not applicable.

**Data Availability Statement:** Data are contained within this article and Supplementary Materials.

**Acknowledgments:** Anonymous reviewers are acknowledged for their constructive comments and helpful suggestions.

**Conflicts of Interest:** The authors declare no conflict of interest.

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
