# Peer review of "Copper and Cadmium Accumulation and Phytorextraction Potential of Native and Cultivated Plants Growing around a Copper Smelter"

_agronomy, doi:10.3390/agronomy13122874_

Round 1

Reviewer 1 Report

Comments and Suggestions for Authors

Comments to the author(s)

Manuscript id.: agronomy-2702202

Title of the manuscript: Copper and cadmium accumulation and phytorextraction potential of native plants growing in a copper smelter

There are some comments, which are to be incorporated in order to improve the manuscript, as given below:

Introduction

*Hypothesis of the work is not well formulated in the ‘Introduction’ section. The authors did not present a novel justification for carrying out this study. What is the hypothesis of the present study?

*It is insufficient and needs more improvement.

*The novelty of the work must be identified and stated more carefully. The authors have to try to explain why this paper is relevant to the wider readership.

*Authors should show the limitations of previous papers.

Results

*Result should be written in concise way.

Discussion

*Need more improvement. It is too superficial and not a meaningful discussion. Author should try to strengthen the discussion part.

*Put reason why this type of result is obtained.

Conclusions

*Authors need to rephrase the "Conclusions" section.

*Add some limitations, underscore the scientific value added of your paper, and/or the applicability of your findings/results and future scope of study.

*Overall, the manuscript needs extensive language editing and formatting. There are so many typography mistakes in the manuscript.

Author Response

We sincerely appreciate your valuable feedback, which we have used to improve the quality of our manuscript. The reviewers' comments are listed below and specific comments have been numbered. Our response is in red text.

Reviewer 2 Report

Comments and Suggestions for Authors

Title

“Copper and Cadmium Accumulation and Phytorextraction Potential of Native Plants Growing in a Copper Smelter”. Did the plants really grow in the copper smelter, or in the area near it, in surroundings?

Abstract

Line 20. “indicating the soils were moderate to heavily pollution level.” Clumsily written sentence, may be – indicating that the level of soil contamination was moderate to severe.

Line 21. were higher than the others?

Line 21-23. No consistency. Why are contents given for Cd, but not for Cu, there are concentrations for the leaves, but not for the roots.

Line 25-26. “The results of TF and BCF for copper and cadmium suggested that the accumulation and translocation capacities for copper were higher than those of cadmium by the eighteen plants.” This is not true; your results show the complete opposite conclusion.

How to understand the last sentence of the Abstract?

Keywords: Repeating the same terms (native plants, phytoextraction) in the title and keywords is not necessary. You can choose others from your manuscript – as different factors.

Introduction

Line 33. Heavy metals (NMs) ?? ?

Line 39. What is soil points? Study sites?

Line 47.” …due to its low cost and environmentally friendly. “Clumsily. May be - because it is inexpensive and environmentally friendly.

Line 58. Did you mean this - Previous studies have shown that wild and cultivated plants, especially hyperaccumulators, were widely used for phytoextraction.

Line 59-61. “For cultivated plants, they may be replaced by local native plant species in the polluted area once human intervention withdrawn.”  Hard to understand what you mean by that? What crops grew in contaminated areas, why did people abandon them?

Line 65. “four plants manually grew around a copper smelter”. What does it mean? Plants or plant species?

The aim of the research should be more clearly formulated, and what do you mean by the word reference.

Materials and Methods

Line 77-78.The soils in the northwest and southeast of the smelter were heavily contaminated, thus the plants were mainly collected from the two directions [17].” From these directions? Why is there a reference here, don't you collected the plants yourself?

Figure 1. “Map of plants with paired soil samples points around Guixi smelter”. It is not a plant map, but a map of plant sample collection sites.

Line 82. “eighteen plants were sampled” – 18 plants or 18 plant species? It is important!

In Table S1, it would be advisable to indicate/highlight which plant species were cultivated.

Line 87. What does it mean – duplicate in triplicate.

Line 88. “The tiny ash in plants surfaces….” – from plant surfaces?

Line 89. Define what you mean by shoot? Did it include leaves too?

Line 95. ground and sieved through 2 mm and 0.15 mm sieves?

The methods must clearly define terms you later used in the Results -what is background soil, geochemical background content in soil, clean soil, reference soil, soil screening values.

Results and Discussion

Line 156-158. It's hard to understand what you're comparing to. What is reference soil in this case?

Line 159-161. This sentence is very vaguely written. Rewrite more clearly.

Line 165-166. … were deleted [13]. Why the reference, don't you process the data yourself?

Line 173-175. If Cu was under the phytotoxic level, how it may be the main reasons for the decline of crop yield and metal accumulation in crops for local area.

Were the Cu contents found in the leaves significantly higher than usually recommended for crops? It would be necessary to discuss it a little and then draw conclusions about the impact on the potential yield. Were there significant differences between species in this respect?

Line 195. “It indicated that the metal accumulation in plants was significantly influenced by total and available metal in soil.” – metal content, concentration?

Figure 3. Correlation coefficients are weak or not visible at all in the Figure. It is necessary to improve the quality of the Figure.

Figure 4 and Figure 5. Abbreviations should also be given in the legends of the Figures (TF, BCF).

Line 247-248. Consequently, we listed the top six plants species which had the potential phytoextraction for copper and cadmium (Table S3). Why is this important summary table in supplementary files? In my opinion, there is a place for it in the article, in addition, precise values of the coefficients could also be given.

Line 258-260. Separate the data from the literature with what you have made clearer in your research. The reference must be after Similar to majority of previous studies.

Line 260. “It is mainly due to the inability to effectively calculate the biomass (kg/ha) of plants in field investigation.” If it was not possible to determine the biomass per hectare, then you could indicate the mass of 1 plant. You also had repetitions. The reader would at least have some idea of the size of the plants of the studied species.

Line 282. “Third, the potential ecological benefits from the phytoextraction process were always neglected.” A very categorical statement. Maybe in this way - often, in most cases...

Line 289. “For instances, Sedum plumbizincicola would be frozen to death under below 273.15 K, and be sunburned to death under above 303.15K.”. How to understand this sentence?

Line 304-306. “Therefore, in the future, we can change the plants with large amount of biomass into fast-growing, valuable, hyper-accumulators through transgenic technology, so as to achieve the purpose of rapidly removing HMs from soil.” Are you completely sure about it? Maybe in terms of desirability though?

Conclusions

Line 310. “The cadmium in the shoot and leaf were found in Sedum plumbizincicola, and the highest cadmium in the root was observed in Phytolacca acinosa Roxb.” -

Did only 1 plant species accumulate Cd in above-ground parts?

In general, the article is interesting, dedicated to such a topical issue as phytoremediation, which is a promising method of remediation of soils contaminated with heavy metals. Suggestions and recommendations for improving the quality of the article are provided in the comments. Although the article is generally written clearly and in good language, there are quite a lot of inaccuracies that need to be corrected and clarified. I suggest making an effort for improving the scientific quality of the present paper. I recommend accepting this article in Agronomy after minor revision.

Author Response

(The authors gave the same response as above.)

Reviewer 3 Report

Comments and Suggestions for Authors

Major comments

1)     Introduction and discussion sections should include the data of previous investigations about soil Cu, Cd phytoremediation based on the utilization of the listed plants. In this respect the authors may use the below references:

 - Xiao-E. Yang, Hong-Yun Peng,Li-Ying Jiang, Zhen-Li He PHYTOEXTRACTION OF COPPER FROM CONTAMINATED SOIL BY ELSHOLTZIA SPLENDENS AS AFFECTED BY EDTA, CITRIC ACID, AND COMPOST Int/J/Phytoremediation 2006, 7 (1) Pages 69-83 https://doi.org/10.1080/16226510590915855

-Xue Z, Wu M, Hu H, Kianpoor Kalkhajeh Y. Cadmium uptake and transfer by Sedum plumbizincicola using EDTA, tea saponin, and citric acid as activators. Int J Phytoremediation. 2021;23(10):1052-1060. doi: 10.1080/15226514.2021.1874290

-Lu, H.; Xu, D.; Kong, T.; Wang, D. Characteristics of Enzyme Activities during Phytoremediation of Cd-Contaminated Soil. Sustainability 2022, 14, 9350. https:// doi.org/10.3390/su14159350

-Yu G, Liu J, Long Y, Chen Z, Sunahara GI, Jiang P, You S, Lin H, Xiao H. Phytoextraction of cadmium-contaminated soils: comparison of plant species and low molecular weight organic acids. Int J Phytoremediation. 2020;22(4):383-391. doi: 10.1080/15226514.2019.1663488.

-iacheng Zou, Fupeng Song, Yanyan Lu, Yuping Zhuge, Yingxin Niu, Yanhong Lou, Hong Pan, Penghui Zhang, Liuying Pang,Phytoremediation potential of wheat intercropped with different densities of Sedum plumbizincicola in soil contaminated with cadmium and zinc,Chemosphere, Volume 276, 2021, 130223,https://doi.org/10.1016/j.chemosphere.2021.130223.

2) In Material and Methods section all plant species investigated should be enumerated with the places of their sampling. In this respect it will be nice if Figure 1 will be presented using bold letters (too difficult to see anything when grey letters are used)

3) It is highly desirable to indicate Families of plant species investigated and give some information about the known frequency of Cd, Cu (hyper) accumulators.

       4) Give some words about the mechanism of Cu and Cd accumulation by plants

Minor comments:

1) Line 4- delete ‘and’

2) Line 54 ‘CEC”- decipher

3) Line 80 ‘Map of plants’ change to ‘map pf locations’

4) Line 155 ‘may due’ change to ‘may be due’

5) Figure 4- what if to compose the diagram placing the x-axis at the level ‘1’???

Author Response

(The authors gave the same response as above.)

Round 2

Reviewer 1 Report

Comments and Suggestions for Authors

The authors have improved the manuscript according to the suggestions. And the manuscript will be accepted in its present form for possible publication.